# Household Splitting Process and Food Security in Malawi

**DOI:** 10.3390/nu15092172

**Published:** 2023-05-02

**Authors:** Maria Sassi

**Affiliations:** Department of Economics and Management, University of Pavia, 27100 Pavia, Italy; maria.sassi@unipv.it; Tel.: +39(0)-382-986-465

**Keywords:** household food security, household splitting process, DID-PSM, Malawi

## Abstract

Despite the frequent changes in household composition in Sub-Saharan Africa, the literature on the household division process is sparse, with no evidence of its effect on food security. This paper addresses the topic in Malawi, where the fission process is evident and malnutrition is a severe problem. Using the Integrated Household Panel Dataset, this study applies the difference-in-difference model with the propensity score matching technique to compare matched groups of households that did and did not split between 2010 and 2013. The results suggest that coping strategies adopted by poor households and life course events determine household fission in Malawi, a process that benefits household food security in the short term. On average, the food consumption score is 3.74 units higher among households that split between 2010 and 2013 compared to the matched households that did not. However, the household division might have long-run adverse effects on food insecurity, especially for poor households due to the adoption of coping strategies that might compromise their human capital and income-generating activities. Therefore, this process warrants attention for the more accurate understanding, design, and evaluation of food security interventions.

## 1. Introduction

This study investigates the effect of household fission on food security in Malawi. In Sub-Saharan Africa, households exhibit frequent changes in their composition due to the evolving nature of life course events and the dynamics of demographic and contextual factors. This phenomenon is also evident in Malawi. However, despite its possible effect on household welfare levels, the process has received less attention in the literature owing to data limitations [1,2,3]. The analysis of household formation is recently possible due to the availability of longitudinal data and specific software for analyzing big data files [4,5]. However, studies focused on Sub-Saharan Africa are still lacking. According to the author’s knowledge, none of them investigates the impact of the household division process on food security. Most studies deal with the effects of rural–urban or between-district migration in the region, considering their impact on different measures of household welfare, such as total consumption [6]. 

In addition, the majority of these studies exclude the household fission that occurs within the administrative area of the origin household. This omission provides an incomplete characterization of the process, and as a consequence its effects on household food security. This aspect is relevant in Sub-Saharan Arica and especially in Malawi, where, for example, marriage is a dominant factor contributing to the household splitting process [7]. Most marriages are formed within the women’s parental district, especially in the matrilineal societies prevalent in about 15 per cent of the Sub-Saharan African ethnic groups [8]. For this reason, the present study focuses on the overall household splitting process in Malawi, irrespective of the destination of the newly generated households. 

Based on the literature [9,10,11], this paper focuses on the overall household fission process in Malawi. This process is defined as a situation in which one or more people resident in an original household leave and form a new split-off household. The study utilizes the two waves of the Malawian Integrated Household Panel Survey 2010–2013–2016–2019 (IHPS), conducted in 2010 and 2013. Based on this dataset, the following typologies of households are considered: the original households, the original non-split households, the original split-off households, and the split-off households. The original households are those that are present in the first wave (2010). These households can generate a split-off household in the second wave (2013) or remain intact [12] (Figure 1). The present study compares the original households in wave 2010 and the original split-off and non-split households in 2013 (light blue shaded area) to detect the effect of the fission process on food security. The household type 1 in Figure 1 is denoted “original household” and in the empirical analysis is part of the treatment group, while the household type 2 is denoted “intact household” and is part of the comparison group.

The empirical analysis in this study compares the 1028 intact and 318 original households in the IHPS dataset to detect and verify whether the household fission process benefits the food security status of original households. However, the comparison is not without challenge, as the households might be systematically different in the characteristics that might determine their food security status, and these characteristics might change over time. As a consequence, the group of original and non-split households might not be comparable. This study addresses this methodological challenge by using a difference-in-difference (DID) method on a matched sample of original and intact households [13], which is obtained using a propensity score matching (PSM) technique [14]. 

Malawi is an ideal country to study the household splitting process and its consequent effect on household food security. The country suffers a significant risk of malnutrition [15], with 51% of the population and 62.9% of households being severely food insecure [16,17]. In this context, the evidence of this paper highlights that the household splitting process results from the factors related to households’ life course events and poverty, which are in turn associated with wealth transfer and resource distribution for the original households. Therefore, the present study aims to verify if this situation produces advantages in terms of a better food security status for the original households compared to the intact households.

This analysis has at least three important aspects. First, the possible effect of fission on household food security contributes to a better understanding of the phenomenon and the impact and design of food security interventions. Failure to account for the unobserved household fission process might overestimate the impact of food security interventions on household food security, especially if the latter is a function of the household division process. Second, it expands the understanding of the overall household splitting process in Malawi and its implications for household food security, solving the limitations of the existing migration literature. Third, it allows for resolving one of the possible selectivity biases in panel data models in the existent empirical literature, where the household structure is assumed to remain constant over time [5].

## 2. Household Splitting Process and Implications for the Original Household

The first step in the empirical analysis was understanding the household splitting process and its potential implications for the food security status of the original household to select the appropriate determinants of household splitting in the econometric model. These determinants were identified based on the literature review, including those described by Piotrowski [18] and key informant interviews. In-person and online interviews were conducted with 12 key informants from international organizations, local NGOs, and INGOs with substantial knowledge and experience of household-level socio-economic and cultural contexts in Malawi. Eight of the informants were female and four were male. The key informants were selected to have different perspectives on the investigated aspects. The interviews lasted 45 min each, and the submitted open-ended questions allowed the informants to speak in-depth. A closing question asked them if they had more to add. At the end of each interview, the content was summarized to ensure that the interviewer and the respondent had the same understanding of the discussion. The interviews were concluded once we found convergence and consensus among the respondents on the determinants of the household splitting process. When possible, the qualitative information gathered from the interviews was triangulated with the literature.

The concept of the household is central to the present study. The household is considered an economic unit; it constitutes a group of people who live together and have a common housekeeping arrangement, which means that they pool their money and eat at least one meal together each day [17]. The study found that the household fission process relates to the realization of life course events and a set of coping mechanisms that poor households implement, as illustrated in Figure 2. This process is associated with wealth transfer and better resource allocation for the original households, with possible positive implications for their food security status.

Regarding the life course events, marriage and the tendency among male youths to seek economic independence were identified as the driving factors behind the household splitting process in Malawi. The country exhibits a pattern of nearly universal marriage, which is the dominant reason for leaving the household and generating a new one [7]. This explanation was corroborated by the responses of the key informant interviews and recent surveys, such as those conducted by the World Bank [17]. The literature also highlights marriage as a social institution that represents the progression of an individual to adulthood as they split from their original home [19].

Almost all marriages in Malawi are contracted under the rules of African customary law. For this reason, there is a strong tie between marriage and land. The association between agricultural livelihood and marriage observed in Sub-Saharan Africa finds confirmation in Malawi [20].

The two forms of customary marriages practiced in the country are patrilineal and matrilineal [21]. In patrilineal societies, bridegrooms pay the bride’s family a bride price called lobola, consisting of money or goods such as small animals. Marriage is also a way to obtain additional land for the original household that commonly forms a part of the dowry [22]. Therefore, lobola represents a transfer of wealth between lineages. 

However, the household splitting process does not occur immediately after the marriage in Malawi. Instead, the key informants revealed that household splitting manifests once the couple gains economic independence, usually after the harvesting season. In the prevalent matrilineal system, the husband moves into the wife’s house, and the wife’s uncle provides the couple with a plot of land, which helps them achieve economic independence after harvesting. Moreover, in matrilineal societies, the chinkhoswe (the engagement ceremony) has become a fundraising function for households over the years. The collected funds provided to the newly wedded couple encourage their economic independence and the formation of a new household.

Similarly, in the patrilineal system, the household splitting process transpires with the achievement of economic independence. However, in this system, the man’s family allocates him the land during childhood, and the bride moves into the groom’s house after marriage. 

More generally, household size reduction following marriage is also seen as a coping mechanism to improve consumption levels in Malawi. However, during the key informant interviews, respondents noted that a possible negative side effect of this mechanism could be a reduced household income if the individual who moved out of the family were a working member. This side effect was mainly associated with the tendency among male youths to seek independence. The literature also asserts this tendency where household fission and the generation of a split-off household is linked to the pursuit of independence, especially among boys at the age of marriage [19].

The present study further identified that household fission in Malawi is also linked to the coping mechanisms employed by poor households. These coping mechanisms include the practice of child marriage and sending members away to form new housing arrangements. 

Malawi is one of the top ten countries with the highest incidence of child marriage in Africa [23]. Approximately 42 per cent of married girls are under the age of 18 and 9 per cent are under 15 [7]. The key drivers of this high rate of child marriage are poverty and the wealth improvements consequent to marriage for the original household [24]. In fact, in Malawi, child girl marriage is supported by traditions and the patriarchal culture. It is considered a means to improve the economic status of the parent families, especially poor families [25]. Kupawila is an example of this cultural practice and its wealth consequences. In this practice, a child’s daughter is offered in marriage to pay off a debt. Another example of this tradition is the Nyakyusa practiced in the Karonga district. According to this tradition, poor and low-status families can exchange their daughters with rich men as collateral for a loan, money, or assets [25]. In addition to income transfer, the household that opts to marry off a daughter benefits from the household size reduction, with possible improvements in resource distribution within the family [26].

The key informant interviews also highlighted another coping strategy that the poorest households implement; they send their members permanently to eat elsewhere and form a separate housekeeping arrangement. However, despite its possible positive effects regarding better within-household resource allocation, our respondents highlighted that this coping mechanism is insufficient to allow the original and newly generated split-off households to escape poverty.

## 3. Materials and Methods

The data used in this study come from the IHPS conducted in Malawi by the National Statistical Office (NSO) as a part of the Integrated Household Survey. The publicly available IHPS dataset covers four waves of a panel survey implemented every three years from 2010 to 2019. Our analysis used the two rounds implemented in 2010 and 2013. This study refers to these survey years as rounds 1 and 2, respectively. It is imperative to note that the present study only used the IHPSs implemented in 2010 and 2013 and omitted those conducted in 2016 and 2019 for the following reasons. 

First, the panel analysis covering the entire survey period between 2010 and 2019 would require consistent data, across all survey years, for all households, including the newly generated split-off households. However, the original households split multiple times during the period 2010–2019, and the split-off households generated in any subsequent survey rounds would not have data for the preceding survey periods. 

Second, Malawi witnessed extreme environmental shocks in 2014–2015 (flood) and 2015–2016 (drought). The environmental shocks themselves could influence the determinants of the household splitting process. Consequently, the effects of the household splitting process on food security observed between the survey periods 2013 and 2016 or 2016 and 2019 might suffer sample selection bias and might not be generalizable over other periods without significant shocks. Therefore, this study focused only on the first two surveys, implemented in 2010 and 2013, to minimize such sample selection bias. Third, the household splitting process between the survey years 2010 and 2016 or 2013 and 2019 was not studied, as households might witness substantial changes in their characteristics in the 6-year gap between these survey periods. 

This paper analyzed the following two mutually exclusive groups of households. The first group comprised original households that generated split-off households between rounds 1 and 2, and the second group included the intact households that did not split during the respective two periods—the intact households. The original household in round 2 was identified by tracking the household head’s unique identification code provided in the dataset. The IHPS panel dataset contained 318 original households, 1028 intact households, and 470 split-off households. The 470 split-off households were dropped in the used dataset, as the present study only concerned the effect of household fission on the original households. The final dataset that was prepared and analyzed contained 318 and 1028 original and intact households with balanced panel data for rounds 1 and 2.

## 4. Variables

Following the literature and the key informant interviews, the present study considered the household splitting process in Malawi as supported by three categories of determinant factors related to the individual-level, household-level, and contextual characteristics of the households. Table 1 presents the descriptive statistics of the outcome variable and the determinants of the household splitting process by household typology and the survey rounds referring to our initial sample before the matching procedure. It only presents the variables used in the final econometric model.

This study estimates the effect of the household splitting process on household food security as measured by the Food Consumption Score (FCS), the outcome variable in this analysis. The FCS is a composite indicator constructed to account for the household dietary diversity, food frequency, and relative nutritional importance of the food groups consumed [27]. The score refers to a 7-day recall period. The FCS was computed following the protocol outlined by the Vulnerability, Analysis, and Mapping Branch of the World Food Programme [28].

The food items of the collected 7-day food frequency data were grouped into the nine standard food groups (main staples, pulses, vegetables, fruit, meat and fish, milk, sugar, oil, and condiments). Afterwards, all consumption frequencies of food items in the same group were summed up and their values were top-coded at 7. Then, the consumption frequency obtained for each food group was multiplied by a standard weight, the value of which depended on the level of energy, quality of protein, and range of micronutrients provided by the food group. These standard weights were 2 for main staples, 3 for pulses, 4 each for meat and fish and milk, 1 each for vegetables and fruit, 0.5 each for oil and sugar, and 0 for condiments. Finally, the weighted food group score for each food category was summed up to determine the household FCS. Households were classified as food-secure if the FCS was above 35, moderately food-insecure if the indicator was between 21 and 35, and severely food-insecure if the FCS was below 21. On average, the original and intact households considered in this study were food-secure in both rounds 1 and 2.

As this indicator is standardized, it can be used to compare the state of households’ food security in different locations and over time [29]. Moreover, validation studies have demonstrated that the FCS is a proxy indicator of household caloric availability [30,31]. However, although the FCS is a good indicator of household food security compared to many other common indicators, it shows some limitations that should be considered, especially when interpreting the results. First, it is a partial measure of food security because it refers only to a short period of time, referring to the 7-day recall period. Therefore, it cannot account for the ‘stability’ dimension of food security in the longer term. Second, the FCS does not provide information on the quantities of food consumed. For this reason, it does not inform on the actual levels of macronutrient intake. Finally, it gives information on household food security but not on the food security and nutritional status of individual members within a household. Therefore, it does not consider possible intrahousehold inequalities (e.g., gender discrimination), which are common in Malawi [32]. 

Regarding the household splitting process determinants, the first variable that captures the individual-level characteristics is the household head’s age expressed in years. In this study, we further added two dummy variables, ‘male (16–25)’ and ‘female (16–25)’, indicating the presence of male and female members in the 16–25 age category. This age range was selected based on the literature and available data to represent the typical age for youths inclined to leave the parental family and generate a new household [33,34,35]. An additional dummy variable was included, ‘female child (8–15)’, with a value equal to 1 if the household had at least one girl child in the 8–15 age category. This study considered only female children because—as previously highlighted—marrying a daughter is a typical coping mechanism used in patrilineal communities to deal with shocks [26]. The lower limit of 8 years of age was chosen as it generally demarcates the girls’ initiation rituals, following which they can marry, in the context of Malawi [7].

This present study represented the household characteristics in terms of its socio-economic status, as proxied by the household size and wealth index. This latter indicator is a widely used composite measure of households’ cumulative living standards in low- and middle-income countries due to difficulties in collecting reliable data on income and expenditure [36]. 

The household size was computed as the total count of members in the household. For the computation of the wealth index, we selected a set of proxy indicators on the households’ ownership of durables and furniture and their dwelling characteristics following the literature [37,38]. The list of durables and furniture selected for the index comprised bicycles, radios, houses, and beds. The dwelling characteristics included improved and unimproved roof and floor materials, access to electricity, the presence of cell phones, the utilization of bed mosquito nets, and household crowding measured in terms of the presence of more than three people per habitable room [39]. These variables were recoded as dummies, and a polychoric principal component analysis was implemented to generate a wealth index as a continuous variable for the analysis, as suggested in the literature [40]. 

The data confirm the evidence from the key informant interviews on the use of the splitting process as a coping mechanism by the poorest household, and that a poor original household generates a poor split-off household. This tendency was verified using household access to social safety net programs targeted toward the poorest and most vulnerable households in Malawi [41,42]. These safety net programs include free maize or other food, public works programs, food or cash-for-work programs, inputs-for-work programs, school feeding programs, targeted nutrition programs, supplementary feeding for malnourished children at a nutritional rehabilitation unit, scholarships or bursaries for secondary education, scholarships for tertiary education, tertiary loan schemes, direct cash transfers from government, and other transfers.

In this paper, it was noted that the original households receiving social safety net programs in both survey rounds tended to generate split-off households in round 2 as new beneficiaries of these programs. Therefore, the study included a dummy variable, ‘social assistance’, with values equal to 1 if the original household received at least one of the institutionally organized household-level social safety net programs in round 2 and 0 otherwise. This variable was used as a proxy indicator of the households’ tendency to utilize coping mechanisms related to sending members permanently away to form separate housekeeping arrangements. 

This study also adopted two dummy indicators for the households’ geographical areas of residence representing the central and southern regions of Malawi to capture the possible confounding factors related to the contextual characteristics of the households. 

## 5. Empirical Strategy

According to the research question of this study, the household splitting process between rounds 1 and 2 was considered as the treatment variable. As described in the previous section, this study hypothesized that the treatment affects household food security, our outcome variable.

The econometric analysis evaluated this hypothesis by referring to the literature on the treatment–effect estimation that ideally requires a randomized assignment of the treatments into two mutually exclusive groups [43]. The groups with and without the treatment are the treated and control groups, corresponding to the original and intact households in our analysis, respectively. 

The implementation of the difference-in-differences (DID) method permitted the treatment effect to be estimated by comparing changes in the household FCS in the two periods analyzed in the treated and control groups [44]. The DID method eliminates the possible confounding bias in the treatment effect in two ways. First, it eliminates the confounding effect of any time-invariant characteristics in the treated and control groups with the pre- and post-treatment comparisons. Second, the DID accounts for the possible confounding effects of time-varying characteristics by differentiating the outcome changes in the treated from those observed in the control groups. 

This study combined the DID method with the propensity score matching (PSM) technique to improve the common trend assumption, as suggested in the literature [44,45]. The PSM method permitted us first to construct the treated and control groups such that each original household in the treated group was matched to an intact household with statistically similar pre-treatment characteristics as in round 1. The study then implemented the DID method within these matched sets of households in the treated and control groups, thereby improving the validity of the common trend assumption.

The study implemented the combined PSM-DID in the following steps, as the literature suggested [46,47]. It first developed the PSM model in (1) to explain the determinants of the household splitting process in a probit regression framework as the following:(1)$$Probit\left(S_{h}\right)=\beta_{0}+\beta_{j}I_{h1}+\beta_{k}H_{h1}+\beta_{l}C_{h1}+\varepsilon_{i1}$$
where *S* is a binary treatment variable, such that its value equals 1 if the household *h* participated in the splitting process between the two rounds and 0 otherwise. Further, *I*, *H,* and *C* represent the set of the households’ individual-level, household-level, and contextual characteristics in round 1, respectively, as presented and discussed in Table 1. Moreover, *β_j_, β_k_*, and *β_l_* are the coefficients of these characteristics, and ε is the residual term.

The predicted probability of *S* in (1) produced an estimated propensity score, which is a composite measure of a household’s pre-treatment characteristics determining its participation in the splitting process [14,46]. These estimated propensity scores were used to match original and intact households that constitute treated and control groups in our analysis, respectively. In the analysis, the households were matched using the kernel function [47]. 

As highlighted in the previous section, this study selected the variables in (1) through literature reviews and key informant interviews. The appropriateness of the PSM model (1) was ascertained by ensuring that the treated and control households fulfilled the common support and balancing conditions. The common support condition requires that households have overlapping propensity scores so that an original household can be matched and compared with an intact household with a similar probability of splitting. The balancing condition requires that the pre-treatment characteristics are, on average, similar both in the overall treated and control groups. The study graphically assessed the common support condition by plotting the propensity scores and asserted the balancing condition using a standardized difference test across the overall treated and control groups. 

Finally, a DID analysis was conducted for the matched treated and control households using the following regression model:(2)$${FCS}_{ht}=\alpha_{0}+\alpha_{1}S_{h}+\alpha_{2}P_{t}+\alpha_{3}{(S}_{h}\times P_{t})+\varepsilon_{it}$$
where *FCS* is the observed food consumption score for household *h* in survey round *t* = {1,2}; *S* is the treatment variable, with values of 1 and 0 for original and intact households h, respectively. Further, *p* is a dummy variable indicating either the pre-treatment (coded as 0) or post-treatment (coded as 1) period, and (*S_h_* × *P_t_*) is the interaction term that informs on the average treatment effect of the splitting process on the *FCS* among households who participated in the splitting process. In technical terms, this treatment effect is referred to as the average treatment effect on the treated (ATT), which is captured by the coefficient α_3_ and is the estimate of our interest.

This study implemented the combined PSM-DID in STATA using a single command ‘diff’, which computed the propensity scores for households, matched original and intact households using a default kernel function, constructed the treated and control groups, and conducted a DID estimation for households fulfilling the common support condition. 

## 6. Results

Table 1 highlights statistically significant differences in characteristics of original and intact households in rounds 1 and 2. These differences signify that the original and intact households are incomparable, and any cross-sectional comparison of their food security levels in either survey round would result in a biased estimation. As described earlier, the present study solved this limitation by employing the PSM technique, which produced comparable groups of treated and control households with a similar propensity to participate in the splitting process. Each household in the treated group refers to the original household that matches the intact household in the control group. 

However, the comparability of the treated and control households depends on the quality and reliability of the matching procedure. Therefore, this study computed the standardized differences in the covariates included in the PSM model (1) to assess the matching procedure, as presented in Table 2. 

Column (ii) of Table 2 reports the percentage of bias for the unmatched sample (U), which was the bias introduced in our analysis when matching procedures were not used. These biases resulted from the systematic difference between the original and intact households, as indicated by the t-test (column (iv)). The results show a considerable reduction in bias for each of the covariates (column (iii)) in the matched sample (M). Consequently, the percentage of bias for all the covariates in the matched sample is significantly lower than the threshold of 5% bias recommended by the literature [13,47]. The *t*-test in the matched sample confirms that the treated and control groups of households are statistically similar after the matching procedure, which ensures their comparability. The estimates also highlight that the treated and control groups of households are balanced in all covariates. This aspect is indicated by the overall covariate balance tests presented in the second section of Table 2, comprising a non-significant likelihood ratio test, a mean bias lower than 5%, and a Rubin’s bias of less than 20 per cent in the matched sample, as recommended by the literature [46,47].

The PSM also identified and evaluated the determinants of the household splitting process and their implications for household food security in Malawi. Therefore, next, the following sub-paragraph presents the probit regression results for the PSM model (1) in Table 3, followed by a discussion on the individual-level, household-level, and contextual determinants of the household splitting process observed in Malawi between the two analyzed rounds.

### 6.1. Determinants of the Household Splitting Process

Table 3 shows that all variables selected for the PSM model (1) are statistically significant and positively associated (except the age squared variable) with the household splitting process. Therefore, they positively affect the predicted probability of household splitting. The statistically significant chi-squared value of the likelihood ratio (LR) test also asserts that the PSM model (1) is overall significant. To ensure that the analysis did not suffer any bias due to the exclusion of any observed determinants, we also combined interactions, higher-order variables, and other demographic indicators in the PSM model (1). However, the literature and the key informant interviews best ascertained the relevance of the determinants in Table 3 in explaining the household splitting process in Malawi. Moreover, these determinants produced a more reliable match of the original and split households than models with other variables. 

Regarding individual-level characteristics, the results suggest that the probability of splitting increases with the increase in the household head’s age. The age squared variable qualifies this process better. Its negative sign indicates that as the household head becomes older, the effect of age on the household division process is lessened. As marriage is the dominant reason for household splitting in Malawi, this non-linear relationship was expected because with the increase in the household head’s age, the household structure changes with the reduction in the number of members in the marriageable age category (16–25). This hypothesis showed a possible confirmation in the dataset used by this study. In fact, there is a negative and statistically significant correlation (−0.43) between the household head’s age and the number of members in the marriageable age category (16–25).

Further, the possible relevance of marriageable members in explaining the household splitting process was also maintained by the positive coefficient of the two specific variables, ‘male (16–25)’ and ‘female (16–25)’. This analysis distinguished between male and female members in the marriageable age category to verify a probable gender component in the splitting process. For this purpose, the t-test of coefficient equality was computed for these dummy indicators. Its *p*-value equaling 0.84 signaled no significant gender differences in the probability of household splitting due to the presence of marriageable male or female members in the household.

The significant and positive coefficient of the ‘female child (8–15)’ indicated that households with an unmarried girl child have a higher propensity to split. This associative effect aligns with the literature that correlates the presence of an unmarried girl child in households and their decision to migrate [48]. The present analysis noted that families with larger household sizes are likely to have more girl children. The household size is also considered a proxy of household-level poverty in the literature. Therefore, the observed effect of the ‘female child (8–15)’ may also signify the role of poverty in the household splitting process and the use of child marriage as a coping mechanism, as suggested by the key informants and the literature [49,50,51].

The positive association of the household size with the household splitting process reinforces this role of poverty in household division. As previously noted, household size is correlated with poverty in Malawi. According to NSO-Malawi [17], 60.6 per cent of households with five or more members were classified as poor in 2019/2020, and almost 28 per cent were ultra-poor. In addition, the chronically poor live in the largest households. Therefore, the results corroborate the literature reports stating that the extreme destitution of these households as the leading cause of their implosion, highlighting their division as a possible effect [52].

The evidence on ‘social assistance’ seems to confirm the use of the household division process by the poorest and most vulnerable households in their attempt to improve their economic conditions. This evidence also sheds light on a mechanism associated with the spread of poverty in the area, despite its possible positive effect on household resource distribution. As previously noted, this group of poor households in the sample tend to generate poor split-off households. However, this coping strategy does not result in their escape from poverty and food insecurity. On the contrary, it might prolong the adverse effects of the household splitting process in the long term.

The positive sign of the estimated wealth index seems to contradict the previous considerations on the role of poverty in the household splitting process. However, the wealth index captures another important aspect of the household splitting process in Malawi, related to the socio-economic status and spouse similarity, as highlighted during our key informant interviews and confirmed by the literature. According to the literature, better educated women are more likely to have better educated husbands, and households with high levels of wealth have more educated members and greater household educational attainment [53,54]. According to the key informants, this assortative mating between educated men and women generally leads to an increased possibility for the couples to become independent and leave the original household, generating a new one. 

The positive association between the two region-specific dummies and the household splitting process affirms the role of the households’ contextual characteristics in their decision to split. In particular, this study highlights that households residing in the central and southern regions are more likely to split than those in the northern regions. These observations are critical in explaining the household splitting process in Malawi, especially considering the heterogeneous cultural and socio-economic contexts across the country’s central, northern, and southern regions. For example, girls in the southern region are observed to marry earlier than those in the northern region, whereas the average marriage age is highest in the central region. In addition, the matrilineal societies dominate the country’s southern region, whereas the patrilineal system prevails in its central areas. The evidence from the present study implies that these contextual characteristics translate into household-level differences, including disparities in the average age of marriage and forms of customary marriages, with specific effects on household wealth [55,56] and their decision to split. 

Furthermore, the regional dummies also captured the varying levels of multidimensional poverty across the three regions, measured along the dimensions of health, population, education, environment, and work. In fact, according to recent estimates, the southern region presents the highest multidimensional poverty index value, the northern region the lowest, and the central region is at an intermediate position [57]. Therefore, the results of this study also provide evidence of the possible positive association between the region-specific destitute living conditions and the household splitting process.

### 6.2. Effect of Household Splitting Process on Household Food Security

Table 4 presents the estimation results for the effect of the household splitting process on household food security for unmatched (U) and matched (M) households in the treated (original households) and control (intact households) groups. 

The first three results in Table 4 were obtained with the pre-treatment and post-treatment comparisons and DID without matching. These results, however, highlight the biased estimated effect of the household splitting process on food security. These estimates are biased due to unobserved differences in the confounding factors (in the pre-treatment and post-treatment comparison) and unreliable common trend assumptions (in the DID analysis conducted without matching). These biased estimation results ascertain the relevance of DID-PSM, which was the methodological approach implemented in our study. 

The DID-PSM estimation result, reported in the final rows of Table 4, is relatively more robust to the biases discussed above. It accounts for both time-invariant and time-variant confounding factors and also improves the plausibility of the common trend assumption due to the application of PSM. 

The DID-PSM indicated a statistically significant increase of 3.74 units in the FCS among the original households in the treated group compared to the non-split households in the control group. This increase corresponds to an average of 7% higher FCS values in 2013 among split households with respect to the base year (2010) net of any increase in the control group. We found that the FCS among the intact households, on average, remained stable across the two analyzed periods. Therefore, any food security improvements observed across these two periods could mostly be attributed to the improvements among original households due to the household splitting process. Such improvements may seem to indicate that the splitting process benefits household food security. However, statistical significance does not necessarily imply practical significance, especially given the lower magnitude of the effect of the splitting process on household food security.

Furthermore, despite any possible beneficial effects, it is also important to consider that the household splitting process might negatively affect some households, especially in the long term. Most of the household splitting process determinants identified in the present study, like child marriage and dependence on social protection systems, are implemented by poor households as coping strategies. These strategies, however, might perpetuate the vicious cycle of poverty and food insecurity for already destitute households. For example, child marriage adversely affects the daughters’ long-term social, economic, and health outcomes due to a lack of schooling, pregnancies, and limited decision-making capabilities at their young age [26]. These effects, therefore, might have far-reaching consequences in terms of further worsening the living conditions for poor families in both original and split-off households, as also suggested in the literature [58].

## 7. Conclusions

The present study confirms the analytical framework presented in Figure 1, which was adopted to investigate the effect of the household splitting process on food security. The process is a manifestation of the household’s circumstantial behavior related to life course events and poverty levels within the Malawian context. The results from the econometric analysis assert that in the short term, the household splitting process marginally improves the food security status for households that split compared to those that do not. 

However, the observed tendency among poor households to generate split-off households as new beneficiaries for the social protection programs might further exacerbate poverty and food insecurity. As the interviews with key informants asserted, such a coping strategy, implemented mainly by poor households, further generates another poor split-off household. At the same time, the strategy could further reduce the human capital in the original household with the exit of income earners. 

Nevertheless, the significant and positive effect of the household splitting process on household food security implies the need to understand the role of this phenomenon in evaluating any food security interventions to avoid a possible confounding bias. The present study’s findings also call for an understanding of the overall household splitting process in the analysis, design, and implementation of food security interventions in Malawi in ways that could help poor households minimize the use of certain coping strategies that might harm them in the long run. These coping strategies include child marriage practices and the tendency of poor households to generate split-off households as beneficiaries of social protection programs.

Future research studies could expand this analysis by quantifying the long-run effects of these coping strategies and the household splitting process on the original and split-off households, along the pathways proposed in our study. Moreover, the methodological approach adopted by this study made an unconfoundedness assumption in the PSM procedure and DID estimation. This assumption entails a similar distribution of unobserved determinants of the household splitting process and household food security situation across the original and intact households. Therefore, future studies could incorporate richer sets of predictors in the econometric models with the possible inclusion of variables indicating intra-household resource distribution and food consumption across both original and intact households. These variables were, however, unavailable in the IHPS dataset. Therefore, the data limitations of this study also warrant the need for future surveys to consider the household splitting process and its consequences in devising questionnaires and data collection tools.

## Figures and Tables

**Figure 1 nutrients-15-02172-f001:**
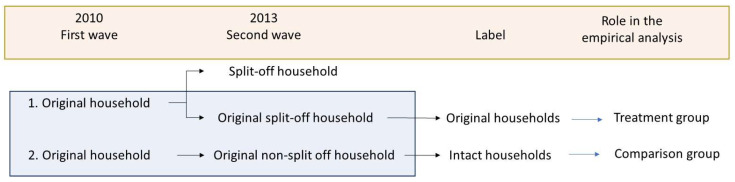
Types of households and labels considered in the empirical analysis. Source: The author’s elaboration.

**Figure 2 nutrients-15-02172-f002:**
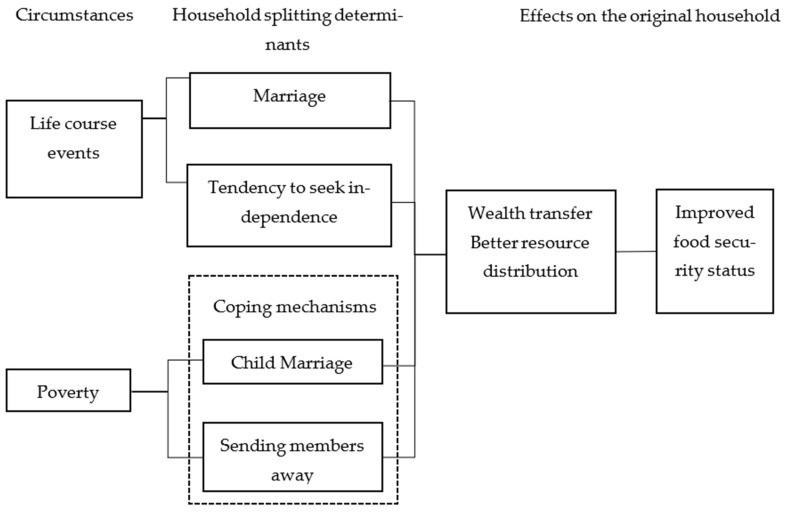
Drivers of the household splitting process. Source: The author’s elaboration.

**Table 1 nutrients-15-02172-t001:** Descriptive statistics for original and intact households before matching.

Variables	Original Households (*n* = 318)		Intact Households (*n* = 1028)		Difference
	M	SD	M	SD	
Round 1 (Year 2010)					
FCS ^1^	51.77	18.42	49.51	18.56	2.261 +
Household head’s age	45.09	14.68	41.31	15.79	3.78 ***
Female (16–25) ^2^	0.51	0.50	0.38	0.49	0.12 ***
Male (16–25) ^3^	0.46	0.50	0.29	0.46	0.17 ***
Female child (8–15) ^4^	0.52	0.50	0.36	0.48	0.16 ***
Household size	5.47	2.27	4.49	2.07	0.98 ***
Wealth index	0.28	1.39	−0.08	1.26	0.36 ***
Social assistance ^5^	0.54	0.50	0.41	0.49	0.13 ***
Region of residence					
Central region	0.46	0.50	0.36	0.48	0.10 **
Southern region	0.49	0.50	0.49	0.50	0.01
Round 2 (Year 2013)					
FCS ^1^	54.79	19.96	50.66	17.36	4.13 **
Household head’s age	49.75	14.21	45.74	16.00	4.01 ***
Female (16–25) ^2^	0.38	0.49	0.35	0.48	0.03
Male (16–25) ^3^	0.40	0.49	0.31	0.46	0.08 *
Female child (8–15) ^4^	0.49	0.50	0.47	0.50	0.02 ***
Household size	4.80	2.22	4.99	2.20	−0.19
Wealth index	0.27	1.36	−0.05	1.30	0.32 **
Region of residence					
Central region	0.50	0.50	0.37	0.48	0.13 ***
Southern region	0.46	0.50	0.48	0.50	−0.02

Note: All statistics are weighted by the sampling weight provided in the IHPS dataset. The households’ region of residence is represented by three dummy variables, with the north region serving as the reference group. ^1^ FCS refers to food consumption score. ^2^ Female (16–25): 1 = a female member is present in the household in the age category (16–25), 0 = otherwise. ^3^ Male (16–25): 1 = a male member is present in the household in the age category (16–25), 0 = otherwise. ^4^ Female child (8–15): 1 = a female child is present in the age category (8–15), 0 = otherwise. ^5^ Social assistance: 1 = household reported to have received social protection programs in 2013, 0 = otherwise. Note: + *p* < 0.1, * *p* < 0.05, ** *p* < 0.01, *** *p* < 0.001.

**Table 2 nutrients-15-02172-t002:** Covariate balance test results for treated (*n* = 318) and control (*n* = 1028) households before and after PSM.

Variables	Mean	Bias%	Reduction in Bias %	t-Statistic
(i)	(ii)	(iii)	(iv)
Treated(Original Households)	Control(Intact Households)			
Household head’s age					
U	44.26	41.11	21.10		3.23 **
M	44.26	43.78	3.20	84.80	0.42
Age Squared × 10^−3^
U	2.16	1.93	16.30		2.46 *
M	2.16	2.12	3.30	79.90	0.44
Female (16–25) ^1^
U	0.53	0.40	25.90		4.06 ***
M	0.53	0.53	−0.30	98.80	−0.04
Male (16-25) ^2^
U	0.47	0.29	37.60		6.02 ***
M	0.47	0.46	2.30	93.90	0.28
Female child (8–15) ^3^
U	0.51	0.35	−32.10		5.07 ***
M	0.51	0.52	−2.80	91.30	−0.34
Household size
U	5.61	4.51	49.40		8.01 ***
M	5.61	5.60	−0.40	99.30	0.04
Wealth index
U	0.44	0.14	21.60	98.30	3.40 **
M	0.44	0.45	−0.40		−0.05
Social assistance ^4^
U	0.51	0.42	18.10		2.84 **
M	0.51	0.49	3.90	78.30	0.49
Region of residence
Central region
U	0.49	0.39	20.00		3.13 **
M	0.49	0.48	1.90	90.50	0.24
Southern region
U	0.45	0.46	−3.30		−0.52
M	0.45	0.45	−1.00	68.60	−0.13
Overall measures of covariate imbalance
Pseudo R-squared	

U	0.10	
M	0.00	
Likelihood ratio test (χ2)	
U	148.35 ***
M	0.92	
Mean bias
U	24.50		
M	1.90		
Median bias			
U	21.30		
M	2.10		

Rubin’s bias	
U	79.5 *		
M	7.6		
Rubin’s R	
U	1.23		
M	1.11		

Note: U: unmatched sample; M: matched sample. All estimates are weighted by the sampling weight provided in the IHPS dataset. The households’ region of residence is represented by three dummy variables, with the north region serving as the reference group. ^1^ Female (16–25): 1 = a female member is present in the household in the age category (16–25), 0 = otherwise. ^2^ Male (16–25): 1 = a male member is present in the household in the age category (16–25), 0 = otherwise. ^3^ Female child (8–15): 1 = a female child is present in the age category (8–15), 0 = otherwise. ^4^ Social assistance: 1 = household reported to have received social protection programs in 2013, 0 = otherwise. Note: * *p* < 0.05, ** *p* < 0.01, *** *p* < 0.001.

**Table 3 nutrients-15-02172-t003:** Probit regression results for determinants of the household splitting process.

Determinants	B	SE
Household head’s age	0.05 **	0.02
Age Squared × 10^−3^	−0.42 *	0.17
Female (16–25) ^1^	0.38 ***	0.11
Male (16–25) ^2^	0.35 **	0.10
Female child (8–15) ^3^	0.23 **	0.11
Household size	0.05 ***	0.03
Wealth index	0.11 **	0.04
Social assistance ^4^	0.23 *	0.10
Region of residence		
Central region	0.89 ***	0.22
Southern region	0.75 *	0.22
Constant	−3.65 ***	0.47
Pseudo R-squared		0.12
LR Test (chi-squared) Total observation (N)		110.49 *** 1346

Note: The estimated results are weighted by the sampling weight provided in the IHPS dataset. The households’ region of residence is represented by three dummy variables, with the north region serving as the reference group. ^1^ Female (16–25): 1 = a female member is present in the household in the age category (16–25), 0 = otherwise. ^2^ Male (16–25): 1 = a male member is present in the household in the age category (16–25), 0 = otherwise. ^3^ Female child (8–15): 1 = a female child is present in the age category (8–15), 0 = otherwise. ^4^ Social assistance: 1 = household reported to have received social protection programs in 2013, 0 = otherwise. Note: * *p* < 0.05, ** *p* < 0.01, *** *p* < 0.001.

**Table 4 nutrients-15-02172-t004:** Estimated effects of the household splitting process on household food security in the treated (*n* = 318) compared to the control (*n* = 1028) households.

Estimation Strategy	Mean FCS				Difference	SE
	Treated (Original Households)		Control (Intact Households)			
	Before	After	Before	After		
Pre-treatment comparison						
U	51.77		49.51		2.26 +	1.32
M	53.40		53.25		0.16	1.09
Post-treatment comparison						
U		54.79		50.66	4.13 *	1.61
M		57.24		53.35	3.90 ***	1.09
DID (without matching)						
U	51.77		49.51		2.26 +	1.32
		54.79		50.66	4.13 *	1.61
	Difference-in-difference				1.87	2.08
DID-PSM						
M	53.40		53.25		0.16	1.09
		57.24		53.35	3.90 ***	1.09
	Difference-in-difference				3.74 *	1.53

Note: U: unmatched sample; M: matched sample. The estimated results are weighted by the sampling weight provided in the IHPS dataset. Note: + *p* <0.1, * *p* <0.05, *** *p* <0.001.

## Data Availability

The data used in this study is publicly available at the Microdata Library of the World Bank.

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
