# Peer review of "Household Splitting Process and Food Security in Malawi"

_nutrients, 2023, doi:10.3390/nu15092172_

Round 1

Reviewer 1 Report

The author of this manuscripts reports a longitudinal study of the relation of household splitting to food security in Malawi.  She analyzed two waves of data from a triennial panel survey.  In addition to finding short-term benefits of household splitting, in this manuscript she provides information about variables that appear to influence household splitting.

The manuscript is well written, the analyses appear to be well executed, and the conclusions appear sound.  My comments are minimal.  There are some points—mostly minor—that I think require serious attention.  The author and editor might want to consider my other points, but I don’t see addressing these as absolutely necessary for publication.

Points that deserve serious attention:

1.     It is somewhat unbelievable that the p value for the difference between household types for Female Child in Round 2 of Table 1 is significant at the .001 level, given some of the other results in that table.

2.     There are various small errors that might be fixed on additional proofreading.  For example, the abbreviation SSA is used in line 130; I assume this means Sub-Saharan Africa, but that abbreviation is not introduced on line 26. (Given that these may be the only two occurrences, it probably shouldn’t be abbreviated on line 130.)

3.     The author says that the PSM method involves matching each original household to a non-treated household.  If this is the case, does this mean that for all of the values in Table 2 on the M (Matched) lines the control sample size was 318?  If so, the column headings in that table that include sample sizes are misleading, and there should be a table note that explains the sample sizes.

4.     Table 2 could probably be made more efficient by not repeating the mean values for the “treated” group on the M and U lines.  Maybe each variable could fit on one line, with columns Treated Mean, Unmatched Control Mean, Matched Control Mean, Unmatched Bias, Matched Bias, Reduction in Bias, Unmatched t-statistic, Matched t-statistic.

5.     The same issues/questions about Table 2 in points 3 and 4, above, apply to Table 4.

6.     In the paragraph on page 12 (lines 451-461) the 3 in parentheses on line 455 should be a 1, and the 3 in parentheses on line 459 should not be in parentheses.

7.     On line 465, “improves” should probably be “increases”.

Points that might be considered:

First, the manuscript is written in first person plural (using, e.g., “we” and “our”).  Given that the manuscript has just a single author, it would read better if written in first person singular (e.g., using “I” and “my”).

Second, the labels for the household types are somewhat confusing, and I suspect that readers would be appreciative of labels that didn’t require as much mental translation and track-keeping and the chosen labels seem to need. The three household types to which the author refers are “original”, “non-split”, and “split-off”.  As I understand things, “original” and “non-split” households were survey respondents in 2010, and in 2013, what was an “original” household in 2010 had lost at least some members (although may have gained some?) by emitting at least one “split-off” household, whereas “non-split” households were still intact.  The problem with this terminology is that the households that do not split by 2013 and are therefore called “non-split” are as original in 2010 as the original households. The classification for 2010 requires looking back from what happened by 2013.  Given that the “split-off” (newly formed) households are never discussed, maybe terms better than “original” and “non-split” would be “Split” and “Intact”, where Split households are households that split between 2010 and 2013 and Intact households are households that remain intact from 2010 to 2013.

Similarly, it is a little confusing for terms that describe household types to be used in parallel with and or interchangeably with the terms “treated” and “control”.  Maybe the existing explanation that “original” households are considered to be “treated” and “non-split” households are considered “control” should be given, after which ideally, the replacement terms for the household types should be used.

A question that might be addressed is whether a household is part of a matrilineal or patrilineal system would be a useful variable, if that information is even available. Would that interact in an important way with the presence of male or female children between 16 and 25?

Although the split-off households are not part of the study, it seems somewhat myopic to not be interested in what happens to them.  Why do we care only about potential benefits for the original households and not potential adversity for the split-off households?  This issue seems to be raised somewhat in the Conclusions, but it might be worth mentioning earlier in the manuscript (e.g., at the point at which the author says that no further attention will be given to the split-off households, around lines 230-233).

The English is fine.  Minor editing would not hurt, but is not required.

Author Response

I appreciate the time and effort you have dedicated to providing valuable feedback on my manuscript. I am grateful for your insightful comments on my paper. I have been able to incorporate changes to reflect all the suggestions you have provided to me. 

Reviewer 1

COMMENT 1: It is somewhat unbelievable that the p value for the difference between household types for Female Child in Round 2 of Table 1 is significant at the .001 level, given some of the other results in that table.

RESPONSE: I ran the STATA command, and the significant level was confirmed.

COMMENT 2: There are various small errors that might be fixed on additional proofreading.  For example, the abbreviation SSA is used in line 130; I assume this means Sub-Saharan Africa, but that abbreviation is not introduced on line 26. (Given that these may be the only two occurrences, it probably shouldn’t be abbreviated on line 130.)

RESPONSE: SSA has been substituted with Sub-Saharan Africa in all cases.

COMMENT 3 to 5:

  1. The author says that the PSM method involves matching each original household to a non-treated household.  If this is the case, does this mean that for all of the values in Table 2 on the M (Matched) lines the control sample size was 318?  If so, the column headings in that table that include sample sizes are misleading, and there should be a table note that explains the sample sizes.

  1. Table 2 could probably be made more efficient by not repeating the mean values for the “treated” group on the M and U lines.  Maybe each variable could fit on one line, with columns Treated Mean, Unmatched Control Mean, Matched Control Mean, Unmatched Bias, Matched Bias, Reduction in Bias, Unmatched t-statistic, Matched t-statistic.

  1. The same issues/questions about Table 2 in points 3 and 4, above, apply to Table 4.

RESPONSE:  the unclear aspect concerning the sample size has been solved, specifying, in the title of tables 2 and 4, the sample size of the control and treated group and removing it from the tables.

If the reviewer accepts, I would like to maintain the structure of the tables because of consistency with the literature. 

COMMENT 6: In the paragraph on page 12 (lines 451-461) the 3 in parentheses on line 455 should be a 1, and the 3 in parentheses on line 459 should not be in parentheses.

RESPONSE: The problem with parentheses has been addressed as sugested.

COMMENT 7: On line 465, “improves” should probably be “increases”.

The word “improves” has been substituted with “increases”.

POINTS TO BE CONSIDERED

COMMENT FIRST: First, the manuscript is written in first person plural (using, e.g., “we” and “our”).  Given that the manuscript has just a single author, it would read better if written in first person singular (e.g., using “I” and “my”).

RESPONSE: The first person plural has been changed using, for example, “this study” or “the present study”.

COMMENT SECOND: Second, the labels for the household types are somewhat confusing, and I suspect that readers would be appreciative of labels that didn’t require as much mental translation and track-keeping and the chosen labels seem to need. The three household types to which the author refers are “original”, “non-split”, and “split-off”.  As I understand things, “original” and “non-split” households were survey respondents in 2010, and in 2013, what was an “original” household in 2010 had lost at least some members (although may have gained some?) by emitting at least one “split-off” household, whereas “non-split” households were still intact.  The problem with this terminology is that the households that do not split by 2013 and are therefore called “non-split” are as original in 2010 as the original households. The classification for 2010 requires looking back from what happened by 2013.  Given that the “split-off” (newly formed) households are never discussed, maybe terms better than “original” and “non-split” would be “Split” and “Intact”, where Split households are households that split between 2010 and 2013 and Intact households are households that remain intact from 2010 to 2013.

Similarly, it is a little confusing for terms that describe household types to be used in parallel with and or interchangeably with the terms “treated” and “control”.  Maybe the existing explanation that “original” households are considered to be “treated” and “non-split” households are considered “control” should be given, after which ideally, the replacement terms for the household types should be used.

RESPOSE: The aspect concerning the labels of household typologies has been first addressed in the introduction. A new figure, Figure 1, clarifies this aspect and the logic of the analysis, introduces the labels used to identify the household groups and associates them with the comparison and treatment group. I maintain the original label, which is consistent with the literature and as explained below. The label non-split-off household, despite being consistent with the language used by the literature, has been changed with intact households to facilitate access to the concept.

These label has been used all across the text.

In the tables, after the treated and control group, I have included in parenthesis the corresponding household label.

COMMENT THIRD: A question that might be addressed is whether a household is part of a matrilineal or patrilineal system would be a useful variable, if that information is even available. Would that interact in an important way with the presence of male or female children between 16 and 25?

Unfortunately, no data is available on the matrilineal or patrilineal systems. 

COMMENT FIRTH: Although the split-off households are not part of the study, it seems somewhat myopic to not be interested in what happens to them.  Why do we care only about potential benefits for the original households and not potential adversity for the split-off households?  This issue seems to be raised somewhat in the Conclusions, but it might be worth mentioning earlier in the manuscript (e.g., at the point at which the author says that no further attention will be given to the split-off households, around lines 230-233).

The new Figure 1 clarifies indirectly why I have not included the split-off households. The comparison group should approximate the counterfactual, representing what the treatment group would have been in case of no treatment. The objective of the study is to follow the original households and verify if their food security changes following the split-off process. It is also for this reason that I have maintained the label “original”.

COMMENT FIFTH: The English is fine.  Minor editing would not hurt, but is not required.

The English language has been proofread.

Reviewer 2 Report

This is an interesting study that examine the impacts of household splitting on food consumption in Malawi. This article is well written and have strong policy implications regarding to food security in the poor regions.  I have the following two comments for improving the manuscript.

1. The author need to consider whether "food security" is an appropriate term used in this study.   The term of "food security" is defined by FAO as whether all people, at all times, have access to sufficient, safe, and nutritious food. As mentioned by the author, FCS provides no information on quantities of food consumed. I am wondering whether FCS can measure "food security".  Probably the term of "diet quality" can be used. 

2. The author need to provide more information on FCS calculation. For example, the author stated that "we multiplied the consumption frequency obtained for each food group by a standard weight whose value depends on the level of energy, quality of protein and the range of micronutrients provided by the food group. " At least, those values for each food group used in this study should be provided in a table or in the supporting information. 

3. I suggest the author to provide a figure to show the distribution of FCSs of all samples, as FCS is an important variable in this study. 

Author Response

I appreciate the time and effort you have dedicated to providing valuable feedback on my manuscript. I am grateful for your insightful comments on my paper. I have been able to incorporate changes to reflect all the suggestions you have provided to me.

COMMENT 1:  The author need to consider whether "food security" is an appropriate term used in this study.   The term of "food security" is defined by FAO as whether all people, at all times, have access to sufficient, safe, and nutritious food. As mentioned by the author, FCS provides no information on quantities of food consumed. I am wondering whether FCS can measure "food security".  Probably the term of "diet quality" can be used. 

RESPONSE: In lines from 260, the study indicates the use of FCS as an indicator of food security following the consolidated literature, which is mentioned in the paper, and introducing the limitations of FCS.

COMMENT 2: The author need to provide more information on FCS calculation. For example, the author stated that "we multiplied the consumption frequency obtained for each food group by a standard weight whose value depends on the level of energy, quality of protein and the range of micronutrients provided by the food group. " At least, those values for each food group used in this study should be provided in a table or in the supporting information. 

RESPONSE: The weights used to compute the FCS have been included in line form 253-255.

COMMENT 3: I suggest the author to provide a figure to show the distribution of FCSs of all samples, as FCS is an important variable in this study. 

RESPONSE: I included the classification of households according to the standard thresholds in lines 256-257. These thresholds have been used to explain the food security status of original and intact households in rounds 1 and 2, in lines from 259, information which is the relevant for this type of analysis.

Round 2

Reviewer 2 Report

The authors have addressed my comments